# Conditions That Simulate the Environment of Atopic Dermatitis Enhance Susceptibility of Human Keratinocytes to Vaccinia Virus

**DOI:** 10.3390/cells11081337

**Published:** 2022-04-14

**Authors:** Matthew G. Brewer, Stephanie R. Monticelli, Mary C. Moran, Benjamin L. Miller, Lisa A. Beck, Brian M. Ward

**Affiliations:** 1Department of Dermatology, University of Rochester, Rochester, NY 14642, USA; benjamin_miller@urmc.rochester.edu (B.L.M.); lisa_beck@urmc.rochester.edu (L.A.B.); 2Department of Microbiology and Immunology, University of Rochester, Rochester, NY 14642, USA; monticellistephanie@gmail.com (S.R.M.); mary_moran@urmc.rochester.edu (M.C.M.)

**Keywords:** keratinocytes, vaccinia virus, atopic dermatitis, barrier disruption, type 2 cytokines

## Abstract

Individuals with underlying chronic skin conditions, notably atopic dermatitis (AD), are disproportionately affected by infections from members of the herpesviridae, papovaviridae, and poxviridae families. Many patients with AD experience recurrent, widespread cutaneous viral infections that can lead to viremia, serious organ complications, and even death. Little is known about how the type 2 inflammatory environment observed in the skin of AD patients impacts the susceptibility of epidermal cells (keratinocytes) to viral pathogens. Herein, we studied the susceptibility of keratinocytes to the prototypical poxvirus, vaccinia virus (VV)—the causative agent of eczema vaccinatum—under conditions that simulate the epidermal environment observed in AD. Treatment of keratinocytes with type 2 cytokines (IL-4 and -13) to simulate the inflammatory environment or a tight junction disrupting peptide to mirror the barrier disruption observed in AD patients, resulted in a differentiation-dependent increase in susceptibility to VV. Furthermore, pan JAK inhibition was able to diminish the VV susceptibility occurring in keratinocytes exposed to type 2 cytokines. We propose that in AD, the increased viral susceptibility of keratinocytes leads to enhanced virus production in the skin, which contributes to the rampant dissemination and pathology seen within patients.

## 1. Introduction

Atopic dermatitis (AD) is the most prevalent inflammatory skin disease, affecting up to 20% of adolescents and 3% of adults worldwide [1]. Patients with AD have increased susceptibility to cutaneous infections from bacterial and viral pathogens. The most notable of these include *Staphylococcus aureus*, herpesviruses (HSV-1 and 2), coxsackieviruses (strains A6 and A16), and poxviruses (molluscum contagiosum and vaccinia virus (VV)) [2,3,4,5]. Some of these cutaneous viral infections can become systemic, resulting in serious morbidity and even mortality. These occurrences are referred to as eczema herpeticum (HSV-1), eczema coxsackium (coxsackieviruses), and the most serious being eczema vaccinatum (VV) [6]. Eczema vaccinatum can lead to prolonged hospitalization and death, with a fatality rate between 5 and 40% [4]. This precludes AD patients from receiving the highly efficacious smallpox vaccine ACAM2000^®^, which contains live replicating VV [7]. While smallpox vaccination for the general public has been discontinued, military personnel (based on deployment location) and laboratory workers with exposure to poxviruses are advised to receive it [8]. Additionally, there are increasing concerns of a clandestine release of smallpox or a zoonotic orthopoxvirus outbreak, such as monkeypox. For all of these reasons, it is critical that we understand which host factors enhance the viral load, with a special focus on AD patients. Since multiple alterations are observed in the uppermost skin layer (i.e., the epidermis) of AD patients, and this is the first portal of contact for invading pathogens, we chose to investigate AD-specific factors that we hypothesize alter the susceptibility of epidermal cells to viral infection.

Several in vitro systems, including primary and immortalized keratinocytes (from healthy or AD patients), as well as murine models that simulate AD, have been developed to better understand why only AD patients develop eczema vaccinatum [9,10,11,12]. Importantly, the dysregulated immune response observed in AD skin has been identified to be a key factor leading to uncontrolled viral infections [13,14]. In addition, recent publications have reported that alterations to the main epidermal cell type, keratinocytes, facilitates viral persistence and dissemination [15,16]. These studies suggest that increased pathogen replication and spread can occur in the skin of AD patients before the adaptive immune response has time to react. Therefore, determining whether AD-specific characteristics (type 2 inflammation and barrier disruption) enhance keratinocyte susceptibility to viral infections will provide new insights into the pathogenesis of disseminated viral infections seen uniquely in this population [17,18,19,20].

The smallpox vaccine is considered one of the most successful vaccines in history, as it led to the eradication of naturally occurring smallpox. VV, the replicating virus in the vaccine, is the prototypical poxvirus and has up to 96% identity to variola virus, the causative agent of smallpox [21]. VV has a broad tropism and is believed to enter most cells [22]. This is thought to reflect either a ubiquitous receptor or the ability to utilize different surface molecules for binding and entry. Different cell types have been shown to produce varying levels of progeny virions, suggesting that variation of the intracellular environment can have dramatic outcomes on virion production [11]. In diseases such as AD, where the skin environment is fundamentally different from healthy skin, the viral replication cycle in keratinocytes could be altered, resulting in enhanced infection and spread.

Most studies focused on epidermal biology utilize primary keratinocytes isolated from either neonatal or adult tissue. Unfortunately, substantial donor-to-donor variability is commonly observed with primary keratinocyte cultures, which can be attributed to genetic variability, host comorbidities, medication use, and post-removal tissue handling. This limitation of primary cell sources can be overcome by using immortalized human keratinocytes. We and others have demonstrated that the immortalized human keratinocyte cell line (N/TERT2G) behaves similarly to primary cells in a variety of assays, including barrier formation, differentiation, stratification, and viral susceptibility [23,24]. With this cell model, we investigated whether type 2 cytokines (IL-4 and IL-13) or barrier disruption alter VV infection of keratinocytes. Furthermore, we validated key findings in primary human foreskin keratinocytes (PHFK). Our results indicate that both type 2 cytokine signaling and barrier disruption independently increase keratinocyte susceptibility to VV, and one could conjecture this would be observed for other viral pathogens. A better understanding of how epidermal alterations could lead to increased viral replication and enhanced pathology may be the first step to designing therapeutics that would help prevent these infections, rather than simply targeting the virus itself.

## 2. Materials and Methods

### 2.1. Cells and Virus

BSC40 cells were obtained from ATCC and maintained in Dulbecco’s modified Eagle’s medium (DMEM—Life Technologies, Grand Island, NY, USA) supplemented with 10% fetal bovine serum. PHFK were isolated from neonatal foreskins and propagated in keratinocyte serum free media (Life Technologies, Grand Island, NY, USA) supplemented with bovine pituitary extract and epidermal growth factor as previously published [25,26]. N/TERT2G keratinocytes were acquired from Dr. Ellen van den Bogaard at the Radboud University (Nijmegen, The Netherlands) and propagated as previously published [24]. PHFK and N/TERT2G keratinocytes were differentiated after confluency by exposure to DMEM media supplemented with calcium [1.8 mM] [27]. A triple fluorescent VV (TrpV) was used for all experiments. This recombinant virus was provided by Dr. John Connor at Boston University [28].

### 2.2. Cytokines and Other Treatments

IL-4 and IL-13 were purchased from Biolegend and used in all assays at 50 ng/mL. The TJDP that demonstrated the most robust disruption in barrier function of PHFK was purchased and reconstituted as previously published [29]. In all assays, the TJDP was used at 10 µM. Pyridone 6 (EMD Millipore—Darmstadt, Germany) was purchased dissolved in DMSO. Cells were exposed to a range of Pyridone 6 concentrations (0.1 nM to 1 µM) or a similar percentage of DMSO prior to VV infection or viability measurements (WST-1 assay as previously published [29]).

### 2.3. Transepithelial Electrical Resistance (TEER) Measurements

TEER was measured as previously published [29]. Briefly, cells were plated at 75,000 cells per transwell. Two days later, culture media was changed to differentiation media containing cytokines, TJDP, or vehicle control (DMSO/Pluronic F127) with media replacements every two days. TEER was measured daily.

### 2.4. Plaque Assays and Quantification of Viral Replication (Titering/qPCR)

Keratinocytes were grown to confluency in 24 well plates and then differentiated. Cells were treated with either IL-4 and IL-13, TJDP, or the vehicle control during differentiation. PHFK were differentiated for 1, 2, or 3 days, and N/TERT2G were differentiated for 2 days and then infected with TrpV at a multiplicity of infection (MOI) of 0.0001. For Pyridone 6 studies, the JAK inhibitor was added 24 h prior to viral infection. After 72 to 120 h post-infection, the media were removed, and the cells were stained with crystal violet to visualize plaque formation. For viral titering, cells were infected similarly to above, and were harvested by scraping at 24 and 48 h post infection (hpi). Samples were subjected to three freeze–thaw cycles to release VV and disaggregated by sonication for 1 min. Infectious VV was titered by plaque assay on BSC40 cell monolayers. In brief, confluent monolayers of BSC40 cells were infected for 2 h with serially diluted cell lysate, infection media were removed, and then, cells were overlaid with DMEM containing 5% FBS and 0.25% methylcellulose. Infected cells were incubated at 37 °C, and 3 days later, media were removed, and monolayers were stained with crystal violet solution. For viral genome quantification, keratinocytes cells were infected with TrpV at a MOI of 3 plaque forming units (pfu)/cell, incubated for 24 h and then analyzed by qPCR as previously described [30]. High MOI growth curves were done similar to viral titering described above. Confluent monolayers were infected with 3 pfu/cell and harvested 6, 12, 18, and 24 hpi then titered on BSC40 cells.

### 2.5. Viral Binding Assay

The 24-well plates containing differentiated keratinocytes (3 days for PHFK or 2 days for N/TERT2G cells) were infected with TrpV at a MOI of 3 for 1 h at 4 °C. Cells were washed twice with 500 µL of ice-cold PBS to remove unbound virus. Cells and bound virus were removed by scraping, and the amount of bound virus was quantified by qPCR as previously described [30].

### 2.6. Plaque Reduction Assay

The number of infectious virions remaining after exposure to conditioned media removed from keratinocytes (N/TERT2G or PHFK) treated with either IL-4 and IL-13, TJDP, vehicle, or media alone, was assessed as previously published [10]. Briefly, 500 pfu of TrpV was added to 600 µL of conditioned media from keratinocyte cultures or unconditioned differentiation media. The virus and media combination were incubated overnight at 37 °C. The remaining amount of virus was quantified by infection of BSC40 cells and plaque enumeration. The number of remaining pfu was normalized to the results obtained from either unconditioned media or untreated conditioned media samples.

## 3. Results

We recently reported that differentiated keratinocytes are more susceptible to VV infection than undifferentiated keratinocytes [23]. Furthermore, AD patients are known to have elevated levels of the type 2 cytokines, IL-4 and IL-13, in their skin [31], leading to the question of what effect do these cytokines have on keratinocyte responsiveness to pathogens. To determine whether the stage of differentiation alters viral susceptibility of keratinocytes before and after exposure to type 2 cytokines, we treated undifferentiated and differentiated N/TERT2G cells with IL-4 and IL-13 (IL-4/13) and then infected them with a low MOI of VV. Viral infection and spread were assessed by plaque formation, which was visualized by crystal violet staining. As we have previously observed [23], differentiated keratinocytes showed enhanced VV susceptibility compared to undifferentiated keratinocytes (Figure 1A,B). The addition of IL-4/13 only increased the susceptibility of differentiated keratinocytes to VV infection as observed by a profound increase in plaque number and size (Figure 1A). To determine whether this result was due to elevated VV spread, we repeated the experiment and harvested infected cells at 24 and 48 hpi to quantify infectious VV produced using a viral titer assay. VV titer from infected keratinocytes was determined on the VV-permissive BSC40 cell line. Quantification of infectious VV showed that differentiated cells treated with IL-4/13 had a ~30-fold increase in infectious virus production at both 24 and 48 hpi compared to media controls (Figure 1B). Undifferentiated keratinocytes treated with IL-4/13 produced similar amounts of virus compared to media-treated cells at both 24 and 48 hpi (Figure 1B). Notably, undifferentiated and differentiated cells produced similar amounts of infectious virus at 24 hpi (Figure 1B) in the absence of IL-4/13. In contrast, by 48 hpi, differentiated cells had substantially increased VV production (374-fold), while there was no increase in VV production from undifferentiated cells (Figure 1B). This supports the idea that there is minimal cell-to-cell VV spread in undifferentiated keratinocytes, even after exposure to AD-relevant cytokines. Furthermore, keratinocytes become more susceptible to VV infection upon differentiation, and this susceptibility is enhanced by the presence of the AD-relevant cytokines, IL-4 and IL-13.

Another key characteristic of the AD environment is epidermal barrier disruption [17,18]. We have shown that barrier formation, as measured by TEER, is diminished in differentiating PHFK by exposure to a patented protein we refer to as a tight junction disrupting peptide (TJDP) [29]. We first confirmed that TJDP also disrupted barrier formation in N/TERT2G, which was similar to our previously published work with PHFK (Figure 2A). To determine whether type 2 cytokines (IL-4 and IL-13) altered barrier in differentiating N/TERT2G cells, we exposed them to IL-4/13 and assessed TEER over three days of differentiation. A modest but statistically significant drop in TEER occurred in cultures exposed to IL-4/13 compared to control cells, indicating that type 2 cytokines modestly diminish barrier function in these cells (Figure 2A). We next assessed whether treatment of N/TERT2G with the TJDP would enhance the susceptibility of these cells to VV infection. N/TERT2G were treated with TJDP or the vehicle during differentiation and infected with VV two days later. For comparison, cells were also treated with IL-4/13. N/TERT2G cells demonstrated markedly enhanced susceptibility to infection following TJDP exposure as measured by plaque number (Figure 2B). Furthermore, the increase was similar to that seen with IL-4/13 exposure. To understand whether this phenotype was driven by increased viral spread, we infected N/TERT2G cells with a low MOI of VV and assessed the amount of infectious virus produced at 24 and 48 hpi. Strikingly, 24 h after infection, IL-4/13 or TJDP treatment of N/TERT2G increased the viral titer by 48- and 47-fold, respectively, compared to media control (Figure 2C). As infection progressed, these differences became less pronounced at 48 hpi (18-fold; IL-4/13 and 29-fold; TJDP) but still remained significant. Interestingly, in the two experiments in which the vehicle for the TJDP was assessed for viral titer (Figure 2C), the mean values were less than the media controls, suggesting that TJDP treatment increases viral spread in KC even more than IL-4/13 treatment. These results indicate that both type 2 cytokines and treatment with a barrier disrupting peptide enhance viral susceptibility of keratinocytes to VV and increase viral spread within the monolayer.

The enhanced susceptibility of the N/TERT2G keratinocytes seen in the presence of either IL-4/13 or TJDP could be the result of increased virus replication. To test this idea, we measured VV genome replication in keratinocytes treated with type 2 cytokines or TJDP. A high MOI infection was used to assure every cell was infected, eliminating viral spread as a confounder in the assay. At 24 hpi, the number of viral genomes produced was quantified by a sensitive VV-specific qPCR [30]. While IL-4/13 treatment of keratinocytes modestly enhanced viral genome replication across multiple experiments, the difference did not reach significance, and TJDP had no effect (Figure 3A). Next, we used a high MOI growth curve to determine whether keratinocytes treated with either type 2 cytokines or the TJDP produce more infectious VV. Infected cells were harvested at 6, 12, 18, and 24 hpi, and the amount of infectious virus produced at each timepoint was determined by viral titer assay. Starting at 12 hpi, keratinocytes treated with either IL-4/13 or TJDP had a noticeable increase in viral titer compared to their respective controls (media or vehicle, respectively), which reached significance at 18 hpi (Figure 3B). Taken together, these results indicate that keratinocytes exposed to IL-4/13 or TJDP produce more infectious virions, which may, in conjunction with their effects on skin barrier (Figure 2A), explain the increased VV spread observed in low MOI experiments (Figure 2C).

Next, we tested the effect of IL-4/13 and TJDP on VV infection in primary human foreskin keratinocytes (PHFK) to evaluate whether they would reaffirm the findings made in N/TERT2G cells. PHFK were grown to confluency and then differentiated in the presence of IL-4/13 or TJDP. For these assays, a triple fluorescent VV (TrpV) was used that expresses a different fluorescent protein at the three stages of poxvirus gene expression: early, intermediate, and late [28]. This virus produces the fluorescent proteins YFP, mCherry, and TagBFP under the C11R, G8R, and F17R viral gene promoters, which express during the early, intermediate, and late stages of transcription, respectively. Using this virus, we were able to approximate the stage of viral genomic expression in keratinocytes. Importantly, TrpV enables the assessment of viral infection and spread in live cell cultures. Treated PHFK were infected with TrpV at a MOI of 0.0001 and imaged by fluorescence microscopy three days post-infection to visualize the expression of the reporter proteins (Figure 4A). Five days post-infection, the monolayers were stained with crystal violet to visualize and count the plaques (Figure 4B). Similar to N/TERT2G cells, PHFK treated with either IL-4/13 or TJDP demonstrated larger plaques at day 3 (Figure 4A) and significantly increased the numbers at day 5 post-infection (Figure 4B). Importantly, untreated and vehicle-treated cells displayed small foci containing relatively few fluorescent cells, suggesting minimal to no viral spread, which was consistent with a lack of intermediate/late gene expression as shown by the absence of mCherry (Red)/TagBFP (Blue) (Figure 4A). Increased plaque number in cytokine- and TJDP-treated cells was detected across multiple PHFK donors, indicating that this biology is commonly observed, regardless of the inherent heterogeneity of the donors (Figure 4C).

Keratinocytes are known to release antimicrobial compounds and peptides, some of which have been reported to target VV [10]. Since we did not detect robust differences in VV genome replication or production of infectious viral progeny in treated N/TERT2G cells, it is possible that keratinocytes exposed to cytokine treatment and/or barrier disruption are producing fewer antimicrobial compounds, which could explain the enhanced viral spread we observed over the course of infection. To assess whether IL-4/13 or TJDP diminished the activity of any antimicrobial compound that had anti-VV activity, we employed a plaque reduction assay in which 500 pfu of VV was added to conditioned media isolated from cultures of either N/TERT2G or PHFK that were differentiated in the presence of either IL-4/13, TJDP, or their controls. Following overnight incubation, the number of remaining infectious virions was determined by the viral titer assay. While we did observe that VV incubated in keratinocyte conditioned media from either N/TERT2G or PHFK demonstrated a ~40% reduction in infectious VV titer (Appendix A), treatment with either IL-4/13 or TJDP did not significantly impact this reduction (Figure 5A).

Previously, keratinocyte barrier disruption was induced by silencing a TJ-specific protein (claudin-1), which resulted in increased HSV-1 infectivity [32]. A potential interpretation of this result is that HSV-1 entry receptors become more accessible after barrier disruption in keratinocytes. Since neither cytokine exposure nor barrier disruption resulted in diminished secretion of factors that altered the infectivity of VV, we next tested whether changes to viral binding (i.e., increased access and/or expression of surface receptors) may be important in the enhanced susceptibility of keratinocytes exposed to either type 2 cytokines or TJDP. To quantify VV binding, we utilized a qPCR assay (similar to genome replication). Both N/TERT2G and PHFK cultures exposed to either IL-4/13 or TJDP demonstrated no significant alterations in VV binding compared to controls (Figure 5B). We did observe a trend toward higher viral binding in TJDP treated samples, but it never achieved statistical significance. These results suggest that the increased viral susceptibility and spread observed in keratinocytes treated with IL-4/13 or TJDP is not due to diminished production of antimicrobial compounds or increased access to VV receptors.

Cumulatively, our results suggest that type 2 cytokine treatment and barrier disruption significantly enhanced VV spread in keratinocytes, which may be in part due to their effects on TJ function (TEER, Figure 2A), but did not seem to be explained by alterations to a specific step of the viral life cycle. The enhanced VV infection observed in response to either IL-4/13 or barrier disruption may be the consequence of a keratinocyte intrinsic signaling pathway. IL-4 and IL-13 activate keratinocytes by binding to the type 1 and type 2 IL-4 receptors expressed on their cell surface [33]. The signaling through these receptors is partly mediated by the Janus tyrosine kinase (JAK) and signal transducer and activator of transcription (STAT) proteins [34,35]. We hypothesized that a pan JAK inhibitor would partially or completely reverse the enhanced VV susceptibility that develops in response to type 2 cytokine treatment of keratinocytes. To test this, we utilized our low MOI infection model in both N/TERT2G and PHFK. Keratinocytes were induced to differentiate in the presence of IL-4/13. At 24 h prior to infection, the pan JAK inhibitor Pyridone 6 (P6) was added to the media of differentiating cells. Cells were then infected with VV, and infection was assessed via plaque assay. Treatment of cells with IL-4/13 and the vehicle for P6 (DMSO) demonstrated similar plaque number compared to IL-4/13 treatment alone, indicating that DMSO does not alter VV infection in keratinocytes (Figure 6A). We observed a dose-dependent decrease in plaque number from both primary and immortalized keratinocytes treated with P6, demonstrating that inhibition of JAK signaling is able to significantly diminish the increased VV susceptibility of IL-4/13 stimulated keratinocytes (Figure 6B,C). Importantly, this reversal in keratinocyte susceptibility to VV infection did not appear to be a result of altered cellular viability due to P6 exposure as assessed by the WST-1 assay (Appendix A). We also tested whether JAK inhibition diminished viral susceptibility of TJDP treated cells but did not observe any change in plaque number (data not shown). These results suggest that inhibition of the JAK/STAT pathway is sufficient to diminish viral susceptibility induced by stimulation with type 2 cytokines. It also suggests that the mechanism by which IL-4/13 enhances keratinocyte viral susceptibility is mechanistically different than that induced by TJDP.

## 4. Discussion

The skin can be divided into three compartments: the epidermis, the dermis, and the subcutaneous layer. The dermis is mostly populated by fibroblasts, which many studies have demonstrated are easily infected by VV and other orthopoxviruses [36,37,38]. The epidermis, which is composed of mainly keratinocytes, are typically not easily infected by viruses, which is consistent with them providing the first layer of protection against external pathogens. Therefore, the epidermis prevents pathogen access to dermal cells, which ultimately may minimize systemic spread. Our data indicate that AD conditions (type 2 inflammation and barrier disruption) dramatically change the susceptibility of the first line of protection. Additionally, a number of murine studies have highlighted the deleterious effects of type 2 immunity (i.e., increased IL-4 and suppression of NK and CTL responses) in the host response to pox viruses [39,40]. None of these studies have looked at the local effect these cytokines (IL-4 and IL-13) may have at the primary site of infection, the epidermis. Our results indicate that both differentiation of epidermal cells and alterations to the cellular environment, such as a change in the cytokine milieu or barrier disruption, enhance susceptibility to viral infection. Many studies have documented changes to keratinocyte differentiation, epidermal inflammatory environment, and cutaneous barrier function in AD [17,20,41,42]. Therefore, understanding how changes to the epidermal compartment affect pathogen susceptibility is critical. To this end, we have demonstrated that keratinocytes (both primary and immortalized) are more susceptible to VV infection only after differentiation [23]. This likely reflects the remarkable transcriptomic changes that occur during the process of differentiation, where keratinocytes transition from a stem cell-like state focused on proliferation to a state of terminal differentiation and subsequent apoptosis [43,44]. Treatment with IL-4 and IL-13 is known to signal transcriptional changes in keratinocytes through the common receptor subunit IL-4Rα [45]. Omori-Miyake et al. have demonstrated that treatment of mouse keratinocytes or the human keratinocyte line, HaCaT, with type 2 cytokines results in downregulation of multiple markers of differentiation, including keratin-10 (KRT10), keratin-1, desmoglein-1, and desmocollin-1 [46]. This suggests that IL-4 and IL-13 may delay the normal differentiation program. We also observed that treatment of human keratinocytes (N/TERT2G) with IL-13 results in decreased expression of KRT10 (Appendix A). Of note, our previous studies demonstrated that the state of differentiation influences keratinocyte susceptibility to viral infection [23]. One possibility from these observations is that IL-4/13 may alter keratinocyte differentiation and prolong the highly VV-permissive differentiation state.

Viruses have complex life cycles, and it is important to investigate how changes to the epidermal environment can impact the stages of viral infection and replication. Here, we demonstrate that VV infectivity (which could be diminished by antimicrobial agents), binding to the surface of keratinocytes, or viral genome replication, is not altered when keratinocytes are exposed to either type 2 cytokines (IL-4/13) or a TJ disrupting peptide. Previous studies have demonstrated that keratinocyte-derived antimicrobial peptides are viricidal against VV and that the expression of one of these antimicrobial peptides, LL-37, was reduced in AD skin explants upon exposure to VV [10,14]. These studies did not recognize the importance of differentiation, which we have observed to be critical in VV susceptibility [23]. Therefore, our results have expanded on their findings by demonstrating that antimicrobial peptides are not important for VV susceptibility of differentiated keratinocytes exposed to type 2 cytokines or whose TJ were disrupted.

Keratinocytes produced in the stratum basale undergo regulated differentiation to form the multiple layers of the epidermis, starting with the stratum spinosum, moving upward through the stratum granulosum where tight junctions are formed, and culminating in the stratum corneum [47]. There are many publications that align with our results and collectively demonstrate that differentiation states or epidermal strata differ in their responsiveness to various stimulations [48,49,50]. Additionally, studies have indicated that multiple cell types (including keratinocytes, fibroblasts, dermal endothelial cells, and Langerhans cells) within the skin are susceptible to VV, with certain subsets producing substantially more progeny virus [11]. In our studies, we focused on keratinocytes, the most dominant cell type in the outermost layer of the skin, because these cells are most likely to be the first cell type exposed to invading viruses. In keratinocytes, we observed that both type 2 cytokines and barrier disruption modestly enhanced production of infectious VV and greatly enhanced VV spread in infected cultures, indicating that inflammatory-dependent and independent alterations to epidermal cells can converge on the same phenotype. Treatment of undifferentiated keratinocytes with IL-4/13 did not enhance either the infection (plaque formation) or the amount of virus recovered from cells. These results suggest to us that VV, and potentially other cutaneous viruses (HSV, coxsackievirus, human papilloma viruses), ideally replicate in specific epidermal strata and inflammatory environments. Components of this hypothesis have been supported by multiple investigators focused mostly on cytokine alterations to the epidermis either through allergy-inducing treatments or viruses producing specific cytokines [9,51,52]. Going forward, we believe a comprehensive study on keratinocytes treated with cytokines representing other inflammatory profiles, including type 1 (IFNγ) and type 3 (IL-17A and IL-22), would identify whether our findings are unique to type 2 inflammation (IL-4/13) or could be attributed to other inflammatory skin diseases.

Finally, we demonstrate that complete inhibition of JAK signaling in type 2 cytokine treated keratinocytes is able to reverse enhanced susceptibility to VV infection. Multiple JAK inhibitors are either recently approved or in late-stage development for treatment of AD [53]. Since this chronic skin condition demonstrates an altered inflammatory skin environment, it is possible that topical JAK inhibitors could limit or prevent cutaneous viral infections, whereas systemic administration may adversely affect adaptive immune responses to viral infections and increase the risk of systemic spread. However, it is notable that baracitinib treatment has improved outcomes for patients with COVID pneumonia. We would still favor topical application, as it would reduce if not eliminate the risks associated with systemic JAK inhibition seen with tofacitinib [54] and would have its major action on keratinocytes.

Patients with AD demonstrate impaired epidermal barrier function. To model this, we previously developed a TJ disrupting peptide that, when used in studies with PHFK and mice, diminished barrier function [29]. Therefore, to test whether alterations to keratinocyte barrier function alter susceptibility to viral infection, we treated keratinocytes with TJDP. Similar to type 2 cytokine exposure, TJDP treatment of keratinocytes enhanced infection with VV, but unlike type 2 cytokines, this occurrence was not mitigated by the pan JAK inhibitor P6. This suggests that IL-4/13 and TJDP enhancement of VV infection may be working through different cellular pathways.

In summary, we showed that keratinocyte differentiation state in tandem with inflammatory milieu (IL-4/13) or barrier disruption (TJDP treatment) critically alters susceptibility to viral infection. Since the epidermis functions as the primary site of infection for many viruses, alterations to the normal homeostasis of the skin can have dramatic effects on pathogen susceptibility. By comprehensively understanding the ideal environment that a pathogen infects and replicates in, we are better able to identify individuals at heightened risk for cutaneous infections and target pathways that can return disease-affected skin to its natural state of resistance to infection.

## 5. Patents

This work provided data for the following patent: PCT WO2020092015A1, Therapeutic mitigation of epithelial infection.

## Figures and Tables

**Figure 1 cells-11-01337-f001:**
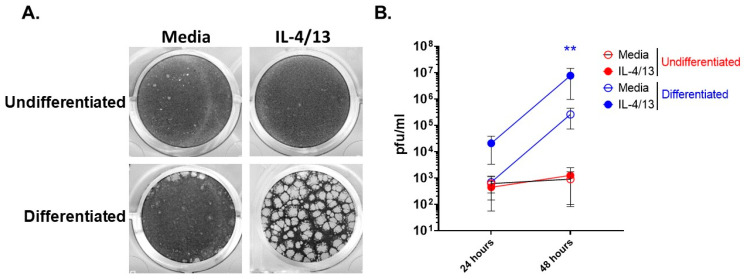
Stimulation of differentiated keratinocytes with type 2 cytokines significantly enhances VV spread. N/TERT2G keratinocytes were either differentiated in Ca^2+^-containing media for 2 days or left undifferentiated ± IL-4/13. Cells were infected with VV at a MOI of 0.0001. Infected monolayers were (**A**) visualized 3 days post-infection with crystal violet or (**B**) harvested at either 24 or 48 hpi, and virus production was quantified by plaque assay. *n* = 3 experiments. Data are shown as mean with standard error of the mean (SEM). Statistical analysis was performed using a paired *t*-test comparing media to IL-4/13 treatment. ** *p* < 0.01, asterisk color signifies the comparison made (e.g., BLUE is for IL-4/13 compared to media from differentiated samples).

**Figure 2 cells-11-01337-f002:**
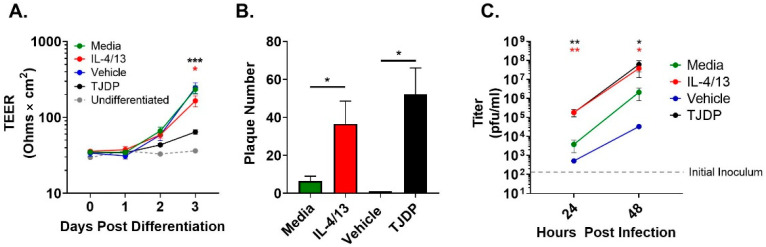
Barrier disruption through TJDP treatment enhances keratinocyte susceptibility to VV infection similarly to type 2 cytokines. N/TERT2G keratinocytes were differentiated in Ca^2+^-containing media in the presence of TJDP, IL-4/13, or DMSO/Pluronic F-127 (vehicle). (**A**) TEER was measured daily and expressed as ohms × cm^2^. *n* = 7 experiments. (**B**) Cells were infected with VV at day 2 post-differentiation with a MOI of 0.0001, and plaques were enumerated 3 days later by crystal violet staining. *n* = 5 experiments. (**C**) Cells infected with VV at a MOI of 0.0001 were harvested at 0 (gray dashed line), 24, and 48 h post-infection, and infectious virus was quantified by plaque assay. *n* = 4 experiments. Data are shown as mean with SEM. Vehicle in (**C**) was assessed in two experiments. Statistical analysis was performed using a paired *t*-test. * *p* < 0.05, ** *p* < 0.01, *** *p* < 0.001 asterisk color signifies the comparison made in C (e.g., RED is for IL-4/13 compared to media, BLACK is for TJDP compared to media).

**Figure 3 cells-11-01337-f003:**
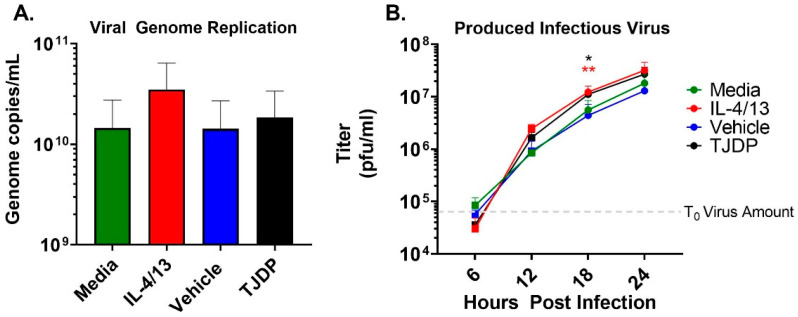
Neither type 2 cytokine or TJDP treatment appreciably increase VV replication in keratinocytes. N/TERT2G were differentiated in Ca^2+^-containing media in the presence of TJDP, IL-4/13 or DMSO/Pluronic F127 (vehicle) for 2 days and then infected with a high MOI of VV. (**A**) At 24 hpi, cells were harvested, and viral genomes were quantified by qPCR. *n* = 4 experiments. (**B**) Treated and infected cells were harvested at 0 (gray dashed line), 6, 12, 18, and 24 hpi, and infectious virus was quantified by plaque assay. Data are combined replicates from 2 experiments. Statistical analysis was performed using a *t*-test comparing media to IL-4/13 or vehicle against TJDP, * *p* < 0.05, ** *p* < 0.01, asterisk color signifies the comparison made in C (e.g., RED is for IL-4/13 compared to media, BLACK is for TJDP compared to media).

**Figure 4 cells-11-01337-f004:**
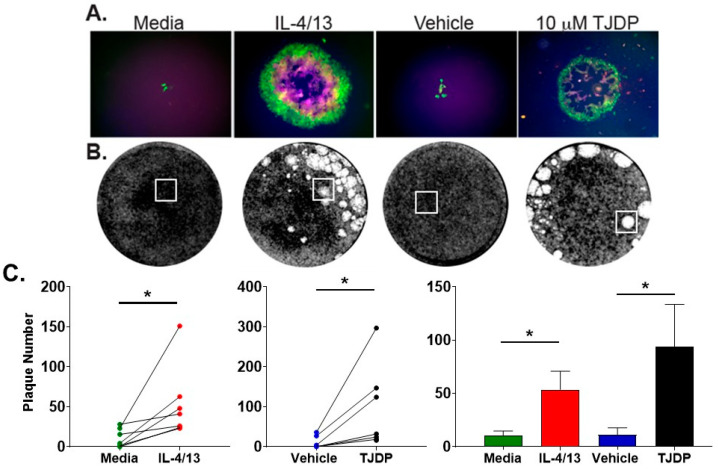
Primary human keratinocytes (PHFK) demonstrate enhanced VV infection after treatment with type 2 cytokines or TJDP. PHFK were differentiated in Ca^2+^-containing media with either TJDP, the vehicle for TJDP, or IL-4/13 for three days before infection (MOI 0.0001). (**A**) Three days post-infection, live monolayers were imaged using fluorescence microscopy for GFP, mCherry, and TagBFP (green, red, and blue, respectively). (**B**) Image of PHFK monolayers stained with crystal violet five days post-infection. White boxes indicate area imaged in the fluorescent panels in (**A**). Images are taken from a single donor. (**C**) Enumeration of plaque number from crystal violet staining. Lines connecting points represent a single donor. Data are shown as mean with SEM from *n* = 7 donors. Statistical analysis was performed using a paired *t*-test comparing media to IL-4/13 treatment and vehicle to TJDP. * *p* < 0.05.

**Figure 5 cells-11-01337-f005:**
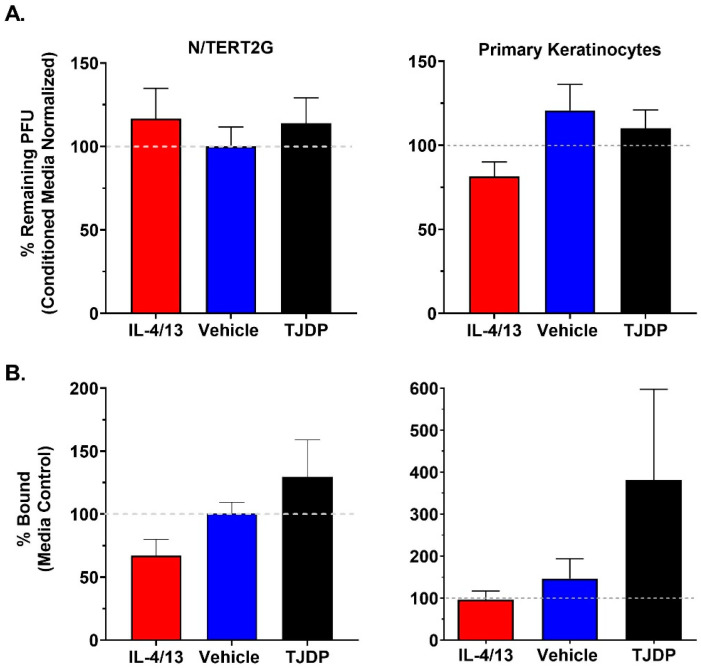
Treatment of KC with type 2 cytokines or TJDP does not alter the antiviral properties of conditioned media or the ability of VV to bind keratinocytes. Keratinocytes were differentiated in the presence of IL-4/13, TJDP, or DMSO/Pluronic F127 (Vehicle). (**A**) Conditioned media were removed from treated cells after two days. VV (500 pfu) was incubated overnight at 37 °C in the conditioned media before quantification via plaque assay. Data are normalized to VV incubated in untreated conditioned media (gray dashed line). N/TERT2G cells; *n* = 4 experiments, PHFK; *n* = 3–7 donors. (**B**) VV was bound to treated keratinocytes on ice for 1 h. Unbound virus was removed by washing, and bound virions were quantified by qPCR. Data are normalized to media control samples (gray dashed line). N/TERT2G; *n* = 4 experiments, PHFK; *n* = 5 donors. Data are presented as mean with SEM. Statistical analysis was performed using a paired ANOVA, comparing samples to media-only controls.

**Figure 6 cells-11-01337-f006:**
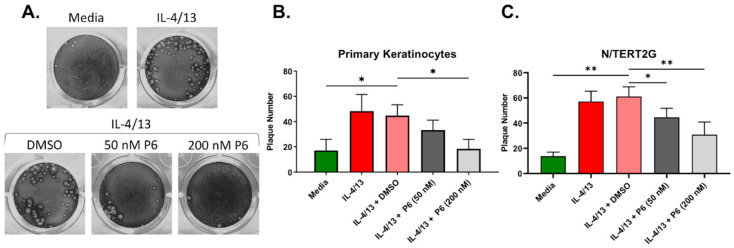
The enhanced VV infection observed in type 2 cytokine treated KC is reversed by pan JAK inhibition. N/TERT2G or PHFK were differentiated in the presence or absence of IL-4/13. One day post exposure to cytokines the pan JAK inhibitor Pyridone 6 (P6) or DMSO (Vehicle) was added. Cells were infected with VV at a low MOI 24 h later. Monolayers were stained with crystal violet 72 hpi to quantify plaque production. (**A**) Representative PHFK monolayers from a single PHFK donor. (**B**,**C**) Quantification of plaque formation in treated PHFK and N/TERT2G. PHFK; *n* = 7 donors, N/TERT2G cells; *n* = 4 experiments. Data are presented as mean with SEM. Statistical analysis was performed using a paired ANOVA, comparing samples to IL-4/13 + DMSO. * *p* < 0.05, ** *p* < 0.01.

## Data Availability

Not applicable.

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
