# Peer review of "Conditions That Simulate the Environment of Atopic Dermatitis Enhance Susceptibility of Human Keratinocytes to Vaccinia Virus"

_cells, 2022, doi:10.3390/cells11081337_

Round 1

Reviewer 1 Report

The study by Brewer at al reports that both type 2 cytokines treatment and barrier disturbance in keratinocytes result in an increased susceptibility to VV. Since these conditions mirror AD phenotype, the authors suggest that these represents critical factors for the VV dissemination and pathology in AD patients. This is well written and well-controlled study that merits publication in Cells. I have several minor concerns that the authors should consider upon revision:

  1. The authors need to include in their discussion and conclusions the data regarding the influence of IL-4 and IL-13 into the differentiation of keratinocytes, published by Omori-Miyake at al., JID, 2014.
  2. The authors demonstrate very well the effect of the Pyridine 6 onto the plaque formation. However, to show that JAK/STAT signaling is blocked in the treated keratinocytes, they should check the phosphorylation status of JAK or STAT6, for example.
  3. The Pyridine 6 is known to inhibits JAK in the low nanomolar range. Why the authors chose to use high concentration in their study?
  4. The size of all figures, especially the diagrams should be increased.

Author Response

Reviewer 1:

The study by Brewer at al reports that both type 2 cytokines treatment and barrier disturbance in keratinocytes result in an increased susceptibility to VV. Since these conditions mirror AD phenotype, the authors suggest that these represents critical factors for the VV dissemination and pathology in AD patients. This is well written and well-controlled study that merits publication in Cells. I have several minor concerns that the authors should consider upon revision:

Comment 1: The authors need to include in their discussion and conclusions the data regarding the influence of IL-4 and IL-13 into the differentiation of keratinocytes, published by Omori-Miyake at al., JID, 2014.

Response: Thank you for your feedback. We have included the Omori-Miyake data in our discussion section (lines 386-390). We have also similarly observed that treatment of human keratinocytes with IL-13 results in decreased expression of keratin-10 suggesting that type 2 cytokines may delay the normal differentiation kinetics in keratinocytes. We have provided this data as supplementary figure 3 in support of the observations from Omori-Miyake et al.

Comment 2: The authors demonstrate very well the effect of the Pyridine 6 onto the plaque formation. However, to show that JAK/STAT signaling is blocked in the treated keratinocytes, they should check the phosphorylation status of JAK or STAT6, for example.

Response: Thank you for this suggestion. We have ongoing studies in which we are testing selective JAK inhibitors, as Pyridone 6 is a pan-JAK inhibitor that inhibits all JAKs. When using specific JAK inhibitors it will be a cleaner system in which we can follow up on the phosphorylation status of specific JAK and STAT molecules. Importantly, we have previously published that Pyridone 6 diminishes STAT3 phosphorylation as a result of IL-17A stimulation [1] albeit at much higher concentrations than used within these studies.

Comment 3: The Pyridone 6 is known to inhibits JAK in the low nanomolar range. Why did the authors choose to use high concentration in their study?

Response: The reported IC50 values were determined by biochemical assays using recombinant proteins [2]. Importantly in cell-based assays the IC50 were observed to shift up to 50-100 nM depending on what cytokine was used for treatment of cells [2]. We have performed a dose response in N/TERT2G with Pyridone 6 and found it to be ineffective at altering IL-4/13 induced viral susceptibility at lower concentrations (<25 nM) in our studies, which is in part shown in Figure 6B where at 50 nM of P6 viral susceptibility is unchanged from IL-4/13 treatment alone.

-The size of all figures, especially the diagrams should be increased.

Thank you for the comment. We have provided high resolution files to the editors so that they can be sized as the journal recommends.

Reviewer 2 Report

In my opinion, the manuscript is very clear, relevant to the field under discussion. It has a typical structure, making it easy to read and draw conclusions. Particularly noteworthy are clear and well presented tables and graphs, illustrating the achieved results. 
    The literature cited is relevant to the subject of the study, includes current papers from various scientific sources. I am very pleased with the subject undertaken by the researchers, because the problem of the influence of infection on the course of AD, and above all, immunological identification of patients susceptible to this phenomenon seems to be a thing of the past in a targeted and individualized therapeutic approach 

Author Response

Reviewer 2:

In my opinion, the manuscript is very clear, relevant to the field under discussion. It has a typical structure, making it easy to read and draw conclusions. Particularly noteworthy are clear and well presented tables and graphs, illustrating the achieved results.

    The literature cited is relevant to the subject of the study, includes current papers from various scientific sources. I am very pleased with the subject undertaken by the researchers, because the problem of the influence of infection on the course of AD, and above all, immunological identification of patients susceptible to this phenomenon seems to be a thing of the past in a targeted and individualized therapeutic approach

Response: Thank you for the comments, we appreciate you taking the time to review our manuscript.